# Modifying emission scenario projections to account for the effects of COVID-19: protocol for CovidMIP

Robin D. Lamboll[1], Chris D. Jones[2], Ragnhild B. Skeie[3], Stephanie Fiedler[4,5], Bjørn H. Samset[3], Nathan P. Gillett[6], Joeri Rogelj[1,7], and Piers M. Forster[8]

[1]Grantham Institute for Climate Change and the Environment, Imperial College London, London, UK
[2]Met Office Hadley Centre, Exeter, UK
[3]CICERO Center for International Climate Research, Oslo, Norway
[4]Institute of Geophysics and Meteorology, University of Cologne, Köln, Germany
[5]Hans-Ertel-Centre for Weather Research, Climate Monitoring and Diagnostics, Bonn/Cologne, Germany
[6]Canadian Centre for Climate Modelling and Analysis, Environment and Climate Change Canada, Victoria, Canada
[7]International Institute for Applied Systems Analysis, Laxenburg, Austria
[8]Priestley International Centre for Climate, University of Leeds, Leeds, UK

**Correspondence:** R. D. Lamboll (r.lamboll@imperial.ac.uk), C. D. Jones (chris.d.jones@metoffice.gov.uk)

**Abstract.** Lockdowns to avoid the spread of COVID-19 have created an unprecedented reduction in human emissions. While the country-level scale of emissions changes can be estimated in near-real-time, the more detailed, gridded emissions estimates that are required to run General Circulation Models (GCM) of the climate will take longer to collect. In this paper we use recorded and projected country-and-sector activity levels to modify gridded predictions from the MESSAGE-GLOBIOM SSP2-4.5 scenario. We provide updated projections for concentrations of greenhouse gases, emissions fields for aerosols and precursors, and the ozone and optical properties that result from this. The codebase to perform similar modifications to other scenarios is also provided.

We outline the means by which these results may be used in a model intercomparison project (CovidMIP) to investigate the impact of national lockdown measures on climate, including regional temperature, precipitation and circulation changes. This includes three strands: an assessment of short-term effects (5-year period), of longer-term effects (30 years) and an investigation into the separate effects of changes in emissions of greenhouse gases and aerosols. This last strand supports possible attribution of observed changes in the climate system, hence these simulations will also form part of the Detection and Attribution Model Intercomparison Project (DAMIP).

## 1   Introduction

Climate change research routinely uses emission scenarios to explore potential future impacts of climate change. These scenarios are developed with Integrated Assessment Models (IAMs) that project internally consistent evolutions of greenhouse gases based on socioeconomic and technological assumptions for the $21^{st}$ century (Weyant, 2017; Riahi et al., 2017; Rogelj et al., 2018). Scenarios are projections, not predictions, and by design reality will differ from the precise evolutions contained in their description. However as we receive more information, the greenhouse gas emission pathways of IAM scenarios can be

modified to more accurately reflect their historical evolution or societal changes. This possibility has gained acute interest in context of the current COVID-19 pandemic.

Societal lockdown measures to contain the spread of COVID-19 have resulted in unprecedented global changes to the emissions of greenhouse gases (GHGs) and aerosols (Le Quéré et al., 2020a; Venter et al., 2020; Forster et al., 2020). There are reports of a 36% reduction in population-averaged global $NO_2$ concentrations (Venter et al., 2020) for 34 countries prior to the 15th of May, and $CO_2$ emissions are estimated to have fallen by 4-8% in 2020 (Le Quéré et al., 2020b; IEA, 2020; Liu et al., 2020; Le Quéré et al., 2021). Shorter-duration and localised changes have been even more extreme (Bauwens et al., 2020; Yang et al., 2020), but show nonlinear changes in air chemistry that simple, globally averaged climate models will miss (Le et al., 2020). Estimates of the immediate impact of this change on global temperature have already been quantified as small using simple climate models (Forster et al., 2020), but these models do not capture complex chemistry, regional temperatures or precipitation effects with any confidence, which are also more uncertain and sensitive to small changes. The unexpected changes in emissions are potentially enough to raise questions about the relevance of projections made only years before. Existing gridded emissions scenarios are poorly designed to even represent realistic emissions changes on less than a ten-year basis.

It is therefore desirable to explore the impact of these changes on climate change projections, both to establish to what extent simulations ignoring the effects so far need updating due to short-term changes and to investigate potential impacts of the lockdown in the long term. This is challenging because country-level emissions estimates are often generated only on a yearly basis, missing the variations between months or weeks. Moreover, detailed climate simulations require emission statistics to be broken down on a higher resolution uniform grid, and these are typically only estimated several years after the emissions have occurred (Feng et al., 2020; Meinshausen et al., 2020).

This paper uses data from near-simultaneous "nowcasting" methods based on open-access data on mobility, energy grids and aviation to modify a pre-existing gridded projection by country- and sector-specific factors. By expressing our scenario as a modification of a pre-existing scenario, we have an estimation of sector emissions on a grid that simulation teams know how to handle. We can also use the pre-existing runs of the baseline scenario as our point of comparison and to provide the initialisation condition for the modified run. This reduces the computational load of running a complete new model when rapid results are desired.

The country-level analysis of Forster et al. (2020) provided a means to assess the level of lockdown affecting different sectors. It used sector-specific changes in Google mobility data, supplemented by an analysis of the legally imposed degree of confinement from Le Quéré et al. (2020a), to produce an up-to-date assessment of the emissions changes in 142 countries. We use updated data following the same technique and add a new methodology for aviation data, then use this to estimate the short-term impact on emissions from COVID-19. The impact of lockdown itself on emissions is likely to be only short-term, since most financial crashes to date have provided only a temporary fall in annual emissions (Le Quéré et al., 2021). However the impact of changes in government investment, towards either the fossil fuel economy or green infrastructure, can have much longer-term impact (Gillingham et al., 2020; Andrijevic et al., 2020). We therefore supplement the short-term emissions modifications with several possible global emissions trends. These diverge from each other only in the future, unlike the pre-

existing SSP scenario sets (which diverge after 2015). This prevents us ascribing any retrospective improvements to the effects of the pandemic.

We use this analysis to generate five scenarios of gridded emissions and concentrations incorporating the effects of lockdown and various different recoveries. We also process the emissions fields through an atmospheric chemistry model to provide the ozone field, often required as an input for General Circulation Models (GCM). We finally describe a protocol for a model

intercomparison project (MIP) assessing the impact of national lockdown measures. Based on discussions between several modelling groups, this activity aims to establish the scope of changes in climate results to be expected from the direct impacts of lockdown, and the potential impact of changes to investment structure resulting from the recovery packages. Since the changes being investigated are likely rather small, the use of a common protocol for modelling groups to perform makes both optimal use of the effort to produce the emissions data and also increases the ability to make robust assessment of the results.

We emphasise the importance of both as large as possible initial condition ensembles and also use of "nudging" techniques for improving the signal-to-noise ratio and for establishing a sufficient body of simulations for the different parts of the MIP.

## 2   Data sources

For this exercise, we change the concentration of the three main greenhouse gases (GHGs): $CO_2$, $CH_4$ and $N_2O$, and emissions of the main aerosols and aerosol and ozone pre-cursors: black carbon (BC), CO, $NH_3$, non-methane volatile organic compounds

(NMVOC) as an aggregate, NOx, organic carbon (OC) and $SO_2$. For aviation emissions, only changes in $CO_2$ and NOx are modelled. Other emissions (such as HFCs) are assumed not to change from their baseline behaviour, since either no change in these emissions is expected or the total impact of these emissions on the climate is so small that a small change in the emissions is likely to have negligible effects.

The baseline data set for our analysis is the MESSAGE-GLOBIOM model SSP2-4.5 Fricko et al. (2017), taken from the

gridded CMIP6 data Input4MIPs Feng et al. (2020). This choice is made for several reasons. Firstly, it is a CMIP6 ScenarioMIP Tier one scenario, meaning that all groups involved in CMIP6 ScenarioMIP have run the baseline scenario O'Neill et al. (2016). Of these, it is the most middle-of-the-road in terms of assumptions, both about future political and economic developments, because it is SSP2 (Riahi et al., 2017) and because it has intermediate long-term forcing, 4.5 $Wm^{-2}$. This amount of forcing is consistent with the global level of warming implied by countries' current NDC pledges (Climate Action Tracker) and has

projected values closest to the most recently measured emissions (Pedersen et al., 2021). SSP2-4.5 is also used in decadal predictions and is therefore of relevance to near term climate forecasts. We emphasise that the code that generates the data that follows can be applied to other scenarios as well.

This baseline scenario is then modified to match the country-and-sector-specific emissions or concentration trends supplied using the methodology of Forster et al. (2020) for times after 2020, updated where such data is available. This technique

projects the emissions change for the most recently measured month to continue at $\frac{2}{3}$ its value, in most scenarios until the end of 2021. In the four year blip scenario we continue this until the end of 2023. In all scenarios we then expect recovery back to baseline over the following year. This is not necessarily the time for the virus to have been completely eliminated or habituated

to, but merely the time for countries to no longer consider lockdowns an effective intervention. After that, we no longer make country-specific modifications but instead modify global emissions by a constant factor, indicating four different styles of recovery from lockdown to have either no difference from baseline (the "two year blip" and "four year blip" scenarios), a transition to an increased use of fossil fuels ("fossil fuel development") or either moderate or large-scale increases in the investment in a green recovery ("moderate green" and "strong green"). The nature of these scenarios is summarised in table 2 and most are described in Forster et al. (2020), although the four year blip is new here. The impact of these paths is felt on different emissions to a different extent (and often with a different sign, for instance greener scenarios emit more $NH_3$ but less $SO_2$), but we do not break down this effect by sector.

As discussed in Forster et al. (2020), data is not available for several regions and sectors, notably including China and all aviation and shipping. In these instances, emissions modifications are taken instead from Le Quéré et al. (2020a), except for aviation emissions, which are instead modified using data obtained from Flightradar24 (2020).

A minor complication of combining these data sources is that the SSP2-4.5 data uses 365-day years, whereas we also have real data from the leap-day in February 2020. To ensure compatibility with climate models, data from the leap day is incorporated into the monthly averages but the output file will not include a day for it.

As this was an evolving project with different amounts of data available at different stages, several versions of the data were released. The details of the code changes involved can be found in table 1.

## 3   Concentration data

Most GCMs use global or hemispherically averaged levels for well-mixed GHGs. These were directly calculated in Forster et al. (2020) using the FaIR v1.5 reduced complexity climate model (Smith et al., 2018). To make this consistent with the general emissions trends found in SSP2-4.5, we calculate the ratio of the concentrations between the baseline and the specific COVID-19 scenarios in the Forster data and apply that multiplier to the global and hemispheric trends in the SSP2-4.5 data to produce the corresponding concentration trends.

A few GCM models use $CO_2$ emissions data, which they put through their own carbon cycle representation. This data is available as described in the Emissions data section below. We remark that the results of the two approaches do not necessarily coincide, since the emissions in Forster et al. (2020) differ from the emissions in this paper in three ways. Firstly, the baseline country emissions in 2020 are based on more recent data than the baseline in SSP2-4.5. Secondly, Forster et al. (2020) uses aviation data based on the Le Quéré et al. (2020a) rather than on more recent Flightradar24 data. Thirdly, Forster et al. (2020) assumes that $CO_2$ emissions from agriculture, forestry and other land use (AFOLU) are reduced by the same amount as the average $CO_2$ emissions change from industry, whereas here we assume no difference in AFOLU emissions. This is due to the finding that global deforestation has not slowed down due to lockdown (Saavedra, 2020; Daly, 2020), and we expect that agricultural output will remain broadly consistent with pre-lockdown levels.

## 4 Emissions data

 ### 4.1 Interpolating additional times

Many gridded IAM models do not report emissions monthly but only on a five or ten-year average basis, and climate models simply interpolate this data for the remaining years. Typically, emissions changes are smooth and the amount of data lost in this way is therefore low. However, when a particularly strong trend occurs suddenly this is difficult to represent on this timescale. Because 2020 is a year normally reported by IAMs, if the emissions for this year were simply corrected without changing anything else then the effects of lockdown would also be felt in the interpolated years before it started, as well as in following years when it is expected to have ended. It is therefore necessary to interpolate additional years onto all datasets with lockdown effects on them – we interpolate 2019, 2021 and 2023. We require data for 2019 to ensure no emissions reduction in the years before lockdown starts. We similarly interpolate 2023 before modifications are made to ensure long-term effects only happen when the model dictates. Since the years 2020 and 2021 are expected to be very different from the surrounding years, they are both interpolated and modified by the effects of lockdown. The year 2022 is defined as exactly equalling the value interpolated, month-for-month, between the effects of lockdown and the baseline behaviour. This is the normal default infilling method of climate simulators so explicit values are not usually needed here. By request from certain groups, monthly data with every year from 2015 to 2025 is available, as is weekly data for 2020 used by Gettelman et al. (2020). Since emissions change on a seasonal basis, interpolated years are interpolated between the same months of the years with available data on either side. This is done before imposing the effects of lockdown, except when we add data for 2022.

### 4.2 Relative emissions factors

The process for handling emissions is more complicated than concentrations, and was subject to a significant change between version 3 and version 4 for shipping. The different versions are described in table 1.

The baseline SSP2-4.5 data contains emissions for nine sectors: AFOLU; energy; industrial processes; surface transportation; residential, commercial and other; solvent production and application; waste; international shipping; and aviation. Aviation emissions are subdivided by altitude, and handled separately. These mostly map well onto the sectors whose activity levels were investigated by Forster et al. (2020), with two exceptions. Firstly Forster et al. model residential and public/commercial buildings separately, so we will use the emissions-weighted mean of these for each country. Secondly, Forster et al. did not have sector-specific estimates for emissions changes from solvents, waste or AFOLU (although $CO_2$ emissions from AFOLU are implicitly assumed to scale with the industrial emissions reduction, as discussed above). We will assume that no changes occurred to these sectors. This is because we do not expect these sectors to be directly affected by lockdown, and the reductions in general economic level will be partly offset by a reduction in regulatory oversight, as has been documented in deforestation changes (Saavedra, 2020; Daly, 2020). We similarly assume that the small island nations and regions like Antarctica not included in the 143 nations estimated by Forster et al. experience no change in emissions, for convenience.

Emissions in the SSP2-4.5 data are broken down by latitude and longitude, so we must classify each emissions pixel as belonging to a single country. We assign each pixel using the reverse_geocoder python module (Thampi, 2015) to the center

of the pixel, which identifies the country that pixel belongs to. It assigns areas of sea to the nearest country. We then check whether the four corners of the pixel are all in the sea using the global_land_mask python module (Karin, 2020). If all four corners are sea, the pixel is instead classified as international waters and is therefore modified by the internationally averaged change in shipping activity rather than the national change in shipping level. Using this definition, only shipping emissions are found in international waters. We emphasise that this classification scheme is purely for emissions calculations and should not be interpreted as a statement of political designation. This treatment of the seas began in version 4 – prior to this, all sea activity used the national shipping activity level of the closest country. Examples of this analysis for April can be found in figure 2, and the globally averaged emissions reduction factors can be found in figure 3. An animation of the global distribution is available in the emissions modification github repository, stored in Zenodo, see Code availability.

## 4.3  Aviation emissions - monthly, versions up to 4

The aviation activity level is always treated globally. The daily number of flights is taken from Flightradar24 free data. This is available from 6/01/2020 up to the time the version is defined. The "null flights" level is calculated as the average number of flights per day in January, and activity level is then expressed as the daily number of flights divided by this. After the end of the available data, we project a linear trend, fitted to data collected after 1/05/2020 (not inclusive), until it reaches the long-term level. This is defined as $\frac{2}{3}$ of the reduction factor of the last complete month of data. In equation form, with angular brackets indicating the mean over the subscript period, $f(t)$ representing flights on the date $t$ days past January 1st and $a(t)$ representing activity level, $f_0 = \langle f \rangle_{Jan}$ and

$$
a(t) = \begin{cases} f(t)/f_0 & \text{if data exists} \\ \min\left(mt + c, 1 - \frac{2}{3}\langle a \rangle_{\text{latest month}}\right) & \text{otherwise} \end{cases}
\tag{1}
$$

for constants $m$ and $c$ that are fit to the data from dates after 1/05/2020. For some versions of the data, the flight activity level is already at the $\frac{2}{3}$ reduction level by the end of the period of collected data so no linear interpolation is seen. The monthly average of this data is then taken to produce the activity level of aviation. This is assumed to be globally uniform and the same across all altitudes. See the graphical illustration in figure 4a.

## 4.4  Aviation emissions - weekly or version 5

Most analyses do not use any higher resolution than monthly, but for one project (Gettelman et al., 2020), weekly data is investigated for the 2020 data. For this project, using open-source data was not required, so we obtained previous years of flight data from FlightRadar24 to better control for seasonal changes. We can then use the weekly-averaged data from 2018 and 2019 for the corresponding day as the baseline instead of the January values:

$$
a(t) = \frac{2\langle f_{2020}(j)\rangle_{j=t:t+7}}{\langle f_{2018}(j) + f_{2019}(j)\rangle_{j=t:t+7}}
\tag{2}
$$

where the subscript on the $f$ indicates the year the flight data is taken from. This produces a weekly-averaged rather than actual daily factor, since it is not possible to decouple seasonal/holiday and weekday effects. Using weekly averages both removes

the weekday effects and reduces the intrinsic variability in the data. This analysis reveals that there are significant seasonal effects, and implies that later versions of the code should also attempt to correct for this - see figure 4b. After the end of the data, we use a linear trend to reach $\frac{2}{3}$ of the last month's average factor as before. As of 08/10/2020, the data for 2019 has also been released open-source, so for version 5 of the data and onwards this approach (with either weekly or monthly averaging) is used for all outputs. We also stop the linear interpolation to baseline for monthly behaviour, since this made little difference to the output with recent results but required the introduction of new parameters to define where the linear trend began.

## 5    Data for aerosol optical properties and associated effects on clouds

Data for the anthropogenic aerosol optical properties and an associated effect on clouds is available via the MACv2-SP parameterisation (Fiedler et al., 2017; Stevens et al., 2017). Models using MACv2-SP can obtain the necessary input data from the supplementary material of (Fiedler et al., submitted) for participating in the CovidMIP experiments. A detailed assessment of the new MACv2-SP data suggest that the global aerosol radiative forcing from CovidMIP will fall within the original spread in the CMIP6 scenarios (Fiedler et al., submitted).

All scenarios from Forster et al. (2020) have been used to create consistent MACv2-SP data (Fiedler et al., submitted). To this end, annual scaling factors for MACv2-SP have been calculated from the $SO_2$ and $NH_3$ emissions from all sectors by following the method previously applied to other gridded emission data from CMIP6 (Fiedler et al., 2019a). The results for the anthropogenic aerosol optical depth, $\tau_a$, point to a global decrease by 10% due to the pandemic in 2020 relative to the baseline. First estimates of the effective radiative forcing associated with anthropogenic aerosols in 2020 point to a less negative global mean by +0.04 $Wm^{-2}$ relative to baseline. Such small ERF differences are difficult to determine due to the large impact of model-internal variability (e.g., Fiedler et al., 2019b). We therefore propose to run ensembles of simulations for participating in CovidMIP. The post-pandemic recovery of emissions is associated with a global $\tau_a$ increase in two out of four scenarios until 2030 and reductions in all scenarios thereafter (Fiedler et al., submitted). In 2050, the $\tau_a$ spread is 0.012 to 0.02. This is a decrease of $\tau_a$ relative to 2005 and relative to four out of nine of the original CMIP6 scenarios for 2050 (Fiedler et al., 2019a). Using the new MACV2-SP data in EC-Earth3 suggests an associated ERF spread of -0.38 to -0.68 $Wm^{-2}$ for 2050 relative to the pre-industrial, which falls within the present-day uncertainty of aerosol radiative forcing (Fiedler et al., submitted).

## 6    Ozone data

For models without ozone chemistry schemes, ozone fields are generated using the OsloCTM3 model (Søvde et al., 2012; Skeie et al., 2020). The OsloCTM3 is a chemistry transport model driven by three-hourly meteorological forecast data by the Open Integrated Forecast System (Open IFS, cycle 38 revision 1) at the European Centre for Medium-Range Weather Forecasts. The horizontal resolution is $2.25° \times 2.25°$ with 60 vertical layers ranging from the surface up to 0.1 hPa. Here, the meteorological data for the year 2014 are used in the simulations.

The emission fields described in section 4 are used as input to the model as monthly fields. Natural emissions including biomass burning emissions are kept constant, and the ozone depleting substances are kept the same in all simulations. The surface methane concentrations are scaled by the increase in concentration since 2019 provided in section 3.

Time slice simulations for the years 2020, 2021, 2023, 2030, 2040 and 2050 are performed using emissions from the four scenarios as well as the baseline scenario SSP2-4.5 (Feng et al., 2020). The ozone in the two year blip scenario is equal to that in the other scenarios for 2020 and 2021 and equal to that of the baseline simulation for the remaining years. The changes in ozone in the two year blip compared to the baseline scenario are shown in Fig. 5a for April 2020 with a decrease of up to 6 percent in the Northern Hemisphere troposphere. Fig. 5b shows the change in total ozone in the different scenarios from 2019 and up to 2050 from the OsloCTM3 simulations relative to the baseline. For 2020 and 2021 the total ozone decreased by 1 Dobson unit (DU) in the two year blip compared to the baseline. In 2023 all scenarios are similar. For the fossil fuel scenario, the ozone changes are positive relative to the baseline scenario but less than 1 DU. The largest change in ozone is for the strong green scenario, with a decrease of 6 DU in 2050 compared to the baseline scenario.

The modelled absolute difference in ozone between the scenario and the baseline are added to the CCMI SSP2-4.5 v1.0 ozone dataset prepared for input4MIPs (Hegglin et al., 2020). The absolute ozone changes are horizontally and vertically interpolated to the same grid as the input4MIPs fields, and the monthly mean values are linearly interpolated for the years in between the years simulated.

## 7    Protocol for CovidMIP

The emissions and concentrations described above are used in CMIP6 Earth system models to simulate the climatic impacts of lockdown. There are three focuses or strands to this MIP. The first is to address the short-term response to the emissions reductions, and the second to address the longer-term response to alternative recovery scenarios. There are sufficient differences in design and groups interested to make this split pragmatic. The third focus is on understanding processes and separating out the role of individual forcing components in contributing to changes in radiative forcing and climate.

Some model groups also have the ability to perform "nudged" simulations which force their model's physical state towards a pre-defined meteorology. This can reduce signal-to-noise issues and help identify aspects of atmospheric composition which might not be apparent in "free running" model simulations. This is preferred where models have this capacity.

It is assumed that model groups have performed the SSP2-4.5 scenario simulations and we use this as a reference set of simulations (baseline) against which we will compare CovidMIP results. Any forcing or aspect of simulation not explicitly defined in this protocol (for example HFCs or land-use) should be kept unchanged from the SSP2-4.5 simulation.

### 7.1    Strand-1. Near-term impact of COVID-lockdown emissions reductions

The goal of these simulations is to assess the impact of COVID-induced lockdown emissions reductions on climate, atmospheric composition and air quality in the near term. To achieve this, we use emissions reductions as close as possible to real emissions as reconstructed from activity data described above. A recovery to baseline emissions is assumed by either (for the

tier 1 emissions case) 2023 or (tier 2) 2025 and simulations should run for 5 years (although longer is also accepted - see section 6.2). For the tier 1 case, we use the two year blip forcing, and for the tier 2 case we use the four year blip forcing. This will be an initial condition ensemble, with model-by-model choice how to arrive at perturbed initial conditions. Note the requirement that parallel SSP2-4.5 simulations already exist, so we anticipate that the same ensemble technique and initial conditions can be used.

Protocol details can be found in table 3.

## 7.2 Strand-2. Longer-term impact of recovery scenarios

This strand uses the three recovery scenarios derived by Forster et al. (2020): strong and moderate green stimulus recovery and a fossil-fuel rebound economic recovery scenario. We place the highest priority (tier 1) on the strong green stimulus recovery as it will likely have the highest signal. As expansion work (tier 3), it includes the short-term lockdown impacts (2 year and 4 year blip scenarios). The experiments are tabulated in table 3. For full details of the scenarios, see Forster et al. (2020), but results are summarised in table 2.

## 7.3 Strand-3. Separation of forcing

COVID lockdown has led to reduced emissions across a wide range of sectors and species. Some of these have competing or offsetting effects on atmospheric composition, radiative forcing and climate. For example, Forster et al (2020) show that at a global level the near term warming due to reduced aerosols may be at least partially offset by reduced greenhouse forcing from ozone. Only on longer timescales does the climate effect of $CO_2$ reductions become significant.

In this strand we use both detection and attribution techniques and fixed-SST diagnosis techniques to isolate and compare the effective radiative forcing (ERF) from individual emission types or categories, and their full implications for regional and global climate evolution.

Two detection and attribution simulations are proposed to parallel ssp245-covid, and allow the separation of the effects of aerosols and well-mixed greenhouse gas perturbations on climate, similar to the way that hist-aer and hist-GHG simulations in DAMIP allow the separation of the effects of these forcings over the full historical period (Gillett et al., 2016). The ssp245-cov-aer simulation is identical to ssp245-covid, except that only aerosol and aerosol precursor emissions (BC, OC, $SO_2$, $SO_4$, $NO_X$, $NH_3$, CO, NMVOCs) follow ssp245-covid, while greenhouse gas concentrations, ozone and all other forcings follow ssp245. Similarly the ssp245-cov-GHG simulation is identical to ssp245-covid, except that only the concentrations of the well-mixed greenhouse gases follow ssp245-covid, while all other forcings follow ssp245. We suggest that groups run as large ensembles of these simulations as possible, but no minimum size is required.

### 7.3.1 ERF calculations

The most commonly used methodology for estimating Effective Radiative Forcing (ERF) is to utilize simulations with fixed sea-surface temperatures (fSST) and prescribed emissions (Richardson et al., 2019; Pincus et al., 2016; Myhre et al., 2013). This

allows the atmospheric conditions to rapidly equilibrate, and rapid adjustments to play out, but broadly avoid the feedbacks associated with a change in surface temperature. For example, (Forster et al., 2016) found thirty years of fSST simulations sufficient to reduce the global 5–95% confidence interval to 0.1 W m$^{-2}$, superior to other methods.

As CovidMIP aims to quantify ERFs that are likely to be relatively weak, on the order of 0.01-0.1 W m$^{-2}$, the recommended protocol is to run 52 year simulations, where the two first years are spinup and the last 50 years are used for analysis. Quantification requires a baseline simulation, and one dedicated simulation for each component to be quantified. Emissions are taken from year 2021 of the "baseline" and "two year blip" scenarios from Forster et al. 2020. For GHG concentrations, we use the prescribed value for 01.01.2021 ($CO_2$, $CH_4$, $N_2O$) for all years. For the SST pattern, we prefer repeated year 2021 values, taken from a coupled simulation, but if this is challenging then another recent year is acceptable so long as the baseline and signal have the same SSTs. Meteorology can vary according to internal variability, but should be representative of year 2021.

CovidMIP defines simulations for diagnosing ERF as follows. For tier 1, to quantify the forcing from aerosols and ozone, we request:

- ssp245-cov-fsst: All emissions from baseline, year 2021

- ssp245-cov-fsst-aer: Aerosol emissions (SOx, BC, OC) from "two year blip", all other emssions from baseline

- ssp245-cov-fsst-ozone: Ozone precursor emissions (NOx, CO, NMVOC) from "two year blip", all other emssions from baseline

For tier 2, to quantify the forcing from individual aerosol species, we request runs with only a single emission species different from the baseline value. The three species for which we want these experiments run are BC, SOx and OC emissions from "two year blip". More details can be found in table 3.

For all strands, we request model groups produce the same diagnostics as per their baseline SSP2-4.5 simulations, reported for the ScenarioMIP.

As an alternative to fSST based ERF diagnosis, some models are able to run nudged simulations where meteorological conditions (typically surface winds and temperatures) are forced to be comparable between signal and baseline. This allows for a direct, time evolving calculation of ERF based on differences in top-of-atmosphere radiative imbalance between the simulations (Chen and Gettelman, 2016; Liu et al., 2018). Although they may not capture the full range of atmospheric adjustments (Forster et al., 2016), nudged ERF calculations are sufficiently comparable to fSST based calculations that they will be used in CovidMIP provided they have prescribed the same emissions as described above.

## 7.4 Anticipated analysis

CovidMIP analysis will primarily start with analysis of two year blip simulations up to 2025. Focus will be on main climate outputs of surface temperature and rainfall, winds and basic circulation and also basic level biogeochemical diagnostics such as carbon stores and fluxes. The first instance of results are available at Jones et al. (2021). These show that the reduction in aerosols is clear in the 2020-2025 period, but the impact of this on temperature or precipitation does not have a clear

enough signal to be detectable in the aggregated data. Some individual models have found clearer trends using nudged analysis (Gettelman et al., 2020).

Similar analysis is planned, but focusing on temperature and precipitation extremes, with analysis based on daily tasmax and precipitation data and a focus on regional aspects. Regional-specific analyses are possible, with East Asia a particular focus region as this is where the largest effects of emissions have been seen in surface aerosols and air quality. The implications of this on local rainfall and monsoon circulation patterns is of particular interest. North Atlantic and European circulation changes will also be investigated.

The effect of emissions reductions on $CO_2$ concentrations is also of interest and may be investigated by ESMs with the capability of performing emissions-driven $CO_2$ simulations. Similarly, ESMs with atmospheric chemistry schemes will be investigated to see the role of emissions reductions on surface ozone and PMs. This will allow us to estimate the global impact of lockdown on health effects.

Model data will be made freely available via the Earth System Grid Federation (ESGF). Users of this data are encouraged to contact model group representatives and invite possible involvement in any resulting publications.

We expect this MIP will allow us to estimate the continued relevance of climate projections that do not include the effects of lockdown. If results significantly deviate from baseline projections, then the continued relevance of outdated simulations is questioned; if results are broadly similar, old projections can be used with more confidence. Initial results indicate that the latter is the case.

## 8   Conclusions

We have demonstrated a novel way to combine data-rich emissions nowcasting with long-term emissions projections to create a dataset suitable for investigating the impact of the large and unforeseen emissions reduction arising from lockdown. This will form the basis for a model intercomparison project to answer questions around how much climatic impact we expect to observe from lockdown measures in both the short and medium term. We also provide ozone fields derived from these results for models that do not produce their own estimates of this. Finally we provide a protocol for how different simulation groups can run experiments on un-initialised, coupled AOGCM/ESM.

## 9   Data availability

The output of these protocols is available from several zenodo addresses. Each address carries the different iterations of the same data where multiple versions are available.

Main set of monthly aerosols emissions and GHG concentrations: https://www.zenodo.org/record/3957826

Four-year blip files for version 4 of the data: https://zenodo.org/record/4446200#.YCKoW-j7SUk

$CO_2$ emissions data, monthly, with data every year 2015-2025: https://zenodo.org/record/3951601

Aerosol emissions data, daily for 2020: https://zenodo.org/record/3952960

NOx emissions from aviation, weekly in 2020: https://zenodo.org/record/3956794

Ozone fields: https://zenodo.org/record/4106460

340 The underlying data for emissions modification terms is available from https://github.com/Priestley-Centre/COVID19_emissions

## 10   Code availability

The code to perform this analysis and to generate an animation of the data is available at

https://zenodo.org/record/4736578.

Old versions of the code and variants can be found at

345 https://github.com/Rlamboll/modify_COVID19_netCDF_Emissions (accessed 19/1/2020).

## 11   Author contributions

CDJ, JR and PMF conceived the project. RDL performed the emissions and concentration analysis, RBS calculated the ozone data and SF provided the MACv2-SP data. CDJ, BHS and NPG determined the protocol for the MIPs. RDL, CDJ, RBS, SF,

350 BHS, NPG and JR contributed to writing the text and all authors were involved in reviewing.

## 12   Competing interests

The authors declare that they have no conflict of interest.

## 13   Acknowledgements

CDJ was supported by funding from the European Union's Horizon 2020 research and innovation programme under grant

355 agreement No 641816 (CRESCENDO). JR, PF, RBS, and RDL were supported by funding from the European Union's Horizon 2020 research and innovation programme under grant agreement No 820829 (CONSTRAIN). CDJ was supported by the Joint UK BEIS/Defra Met Office Hadley Centre Climate Programme (GA01101). SF acknowledges the funding for the Hans-Ertel-Centre for Weather Research from the German Federal Ministry for Transportation and Digital Infrastructure (grant: BMVI/DWD 4818DWDP5A).

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

| Version no. | Data date | Notes |
|---|---|---|
| 1.0 | 14/05/2020 | First available data |
| 3.0 | 17/06/2020 | Major bugfix - data before this point should not be used. |
| 4.0 | 14/07/2020 | Pixels whose four corners are in the sea use internationally averaged shipping factors |
| 5.0 | 25/01/2020 | Substantial update of data to cover 2020. Bugfix affecting aerosol values on December 2021. |

**Table 1.** Table of noteworthy difference between versions of data. The first digit of the version number is incremented by both additional months of complete data and by major coding developments. The second digit represents significant coding changes or additional data use within the same final month of data - these data are broadly intercompatible. The third decimal place denotes changes in the times at which data is reported or minor bugfixes.

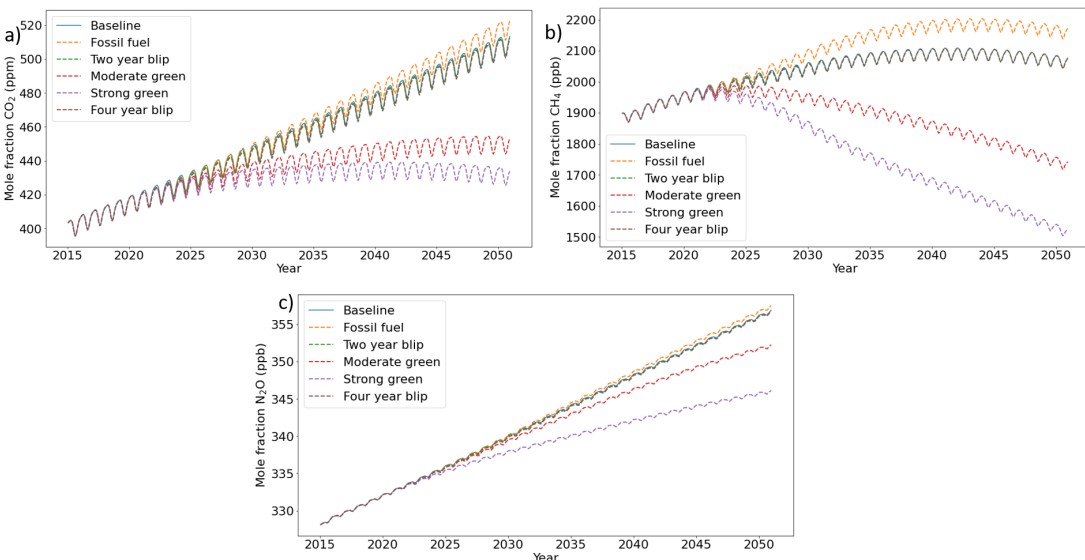

**Figure 1.** Concentrations for the three persistent GHGs, a) $CO_2$, b) $CH_4$, c) $N_2O$. In each case the baseline data is very similar to two and four year blips, hence the difficulty distinguishing the lines.

| Scenario | Assumptions |
| --- | --- |
| Baseline | SSP 2-4.5 data is used without modification. |
| Two year blip | Data is modified for all of 2020 and 2021 in accordance with observed activity levels in the sectors of different countries. This is projected to continue at $\frac{2}{3}$ of the activity reduction value for the latest month available for the rest of the two year period. Activity is interpolated, month-for-month, back towards baseline over 2022 and is equal to baseline thereafter. |
| Four year blip | As for two year blip, except the projected activity reduction, of $\frac{2}{3}$ of the last month available, is continued until 2023. Activity is interpolated, month-for-month, back towards baseline over 2024 and is equal to baseline thereafter. |
| Fossil fuel | Follows two year blip until 2023. Thereafter, the effects of additional investment in fossil fuels during recovery are included in a globally uniform way. Financial modelling produced estimated global Kyoto gas emissions totals consistent with 10% higher emissions than the path met if countries meet their nationally determined contributions (NDCs). We used the open-source package Silicone (Lamboll et al., 2020) to find a linear combination of MESSAGE-GLOBIOM SSP2 scenarios that gave the same total Kyoto emissions. We use the global relative emissions level of each aerosol and precursor in this composite scenario to rescale the 2D emissions maps. The relative concentration change arising from this scenario is used to rescale global greenhouse gas concentrations. |
| Moderate green | Follows two year blip until 2023. Thereafter, the effects of small additional investment in green technology are included in a globally uniform way. Financial considerations as to what emissions change is plausible with moderate ambition (in keeping with results in McCollum et al. (2018)) produced a Kyoto emissions total in 2030 of 35% lower than the NDCs, which we resolve into a linear combination of MESSAGE-GLOBIOM SSP2 scenarios. We then set a global net zero $CO_2$ trajectory for 2060, and resolve this $CO_2$ total into a linear combination of MESAGE-GLOBIOM SSP2 scenarios again using Silicone. The relative difference between this scenario and the baseline is used to rescale emissions and concentrations as in the fossil fuel case. |
| Strong green | Follows two year blip until 2023. Thereafter, the effects of large additional investment in green technology are assumed to push the scenario towards an IMAGE SSP1 world. In 2030 we are assumed to reach the emissions rate of SSP1-19, around 52% lower than following current NDCs, and thereafter follow a global net zero $CO_2$ target for 2050. The other emissions are formed by a linear combination of IMAGE SSP1 scenarios that give the closest total $CO_2$ match to this pathway. (This composite pathway is always close to the SSP1-19 pathway after 2023.) |

**Table 2.** Summary table for the differences between scenarios. For more details on how these were constructed see Forster et al. (2020). More details on the calculation of the emissions values themselves can be found in the file 'InfillingCovidResponse.ipynb' in Lamboll (2020).

| Strand | Name | Tier | Run length | Recommended ensemble size | scenarios |
|--------|------|------|-----------|--------------------------|-----------|
| 1 | ssp245-covid | 1 | 5 years or more | Large as possible, 10+ members | Two year blip |
| 1 | ssp245-covid4yr | 2 | 5 years or more | Large as possible, 10+ members | Four year blip |
| 2 | ssp245-cov-strgreen | 1 | 31 years | 10 members | Strong green |
| 2 | ssp245-cov-modgreen | 2 | 31 years | 10 members | Moderate green |
| 2 | ssp245-cov-fossil | 2 | 31 years | 10 members | Fossil fuel |
| 2 | ssp245-cov-2yr | 3 | 31 years | 10 members | Two year blip |
| 2 | ssp245-cov-4yr | 3 | 31 years | 10 members | Four year blip |
| 3 | ssp245-cov-aer | 1 | 5 years or more | Large as possible, 10+ members | Two year blip for aerosols |
| 3 | ssp245-cov-GHG | 1 | 5 years or more | Large as possible, 10+ members | Two year blip for GHGs |
| 3 | ssp245-cov-fsst | 1 | 52 years | Large as possible, 10+ members | All baseline |
| 3 | ssp245-cov-fsst-aer | 1 | 52 years | Large as possible, 10+ members | Two year blip for SOx, BC and OC |
| 3 | ssp245-cov-fsst-ozone | 1 | 52 years | Large as possible, 10+ members | Two year blip for NOx, CO and NMVOC |
| 3 | ssp245-cov-fsst-bc | 2 | 52 years | Large as possible, 10+ members | Two year blip for BC |
| 3 | ssp245-cov-fsst-sox | 2 | 52 years | Large as possible, 10+ members | Two year blip for SOx |
| 3 | ssp245-cov-fsst-oc | 2 | 52 years | Large as possible, 10+ members | Two year blip for OC |

**Table 3.** Table of experiments for CovidMIP. All experiments branch from SSP2-4.5 on 01/01/2020. Emissions/concentrations not specified come from the baseline data.

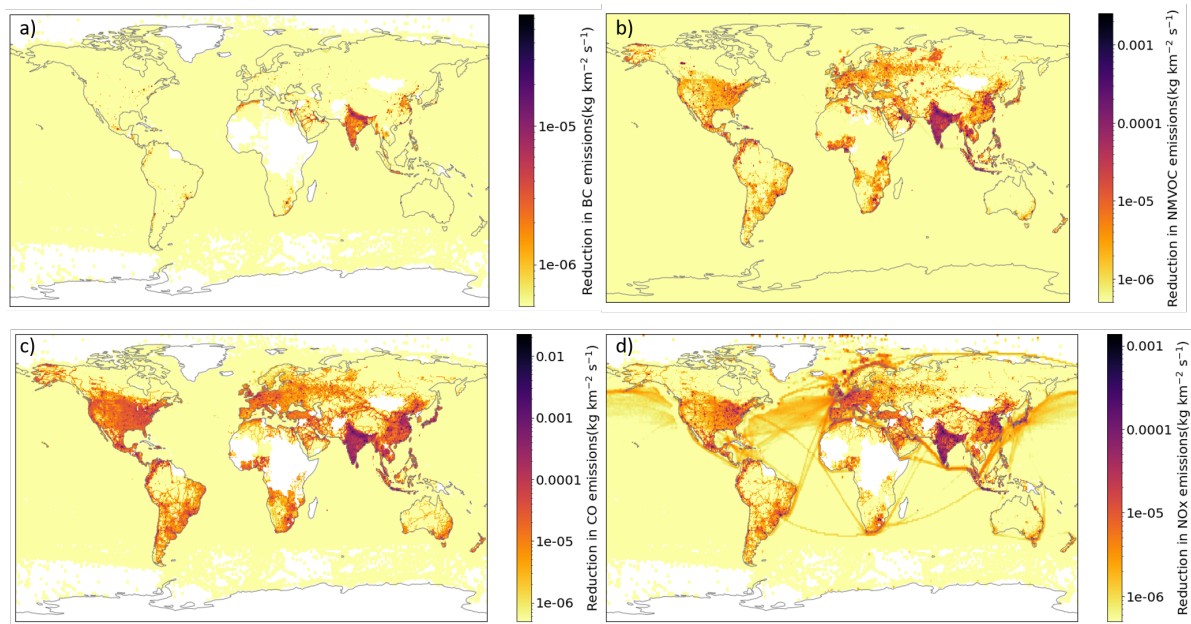

**Figure 2.** Difference in emissions between baseline and the two-year blip COVID-19 scenario during April 2020. White regions indicate that the emissions change was zero (often due to emissions being zero in the first place) or emissions increased. Species are a) BC, b) NMVOC, c) CO and d) NOx.

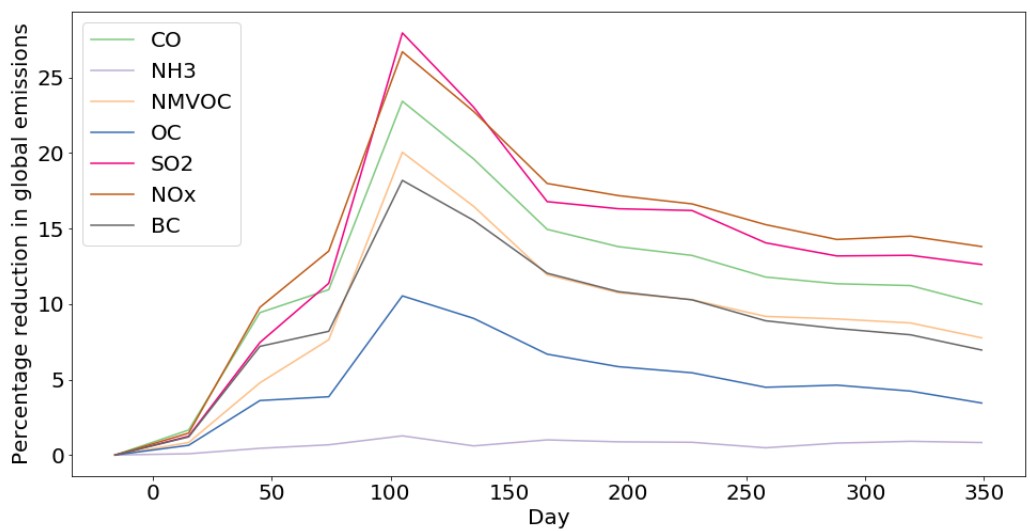

**Figure 3.** Monthly global emissions reduction estimates for 2020 in version 5 of the data.

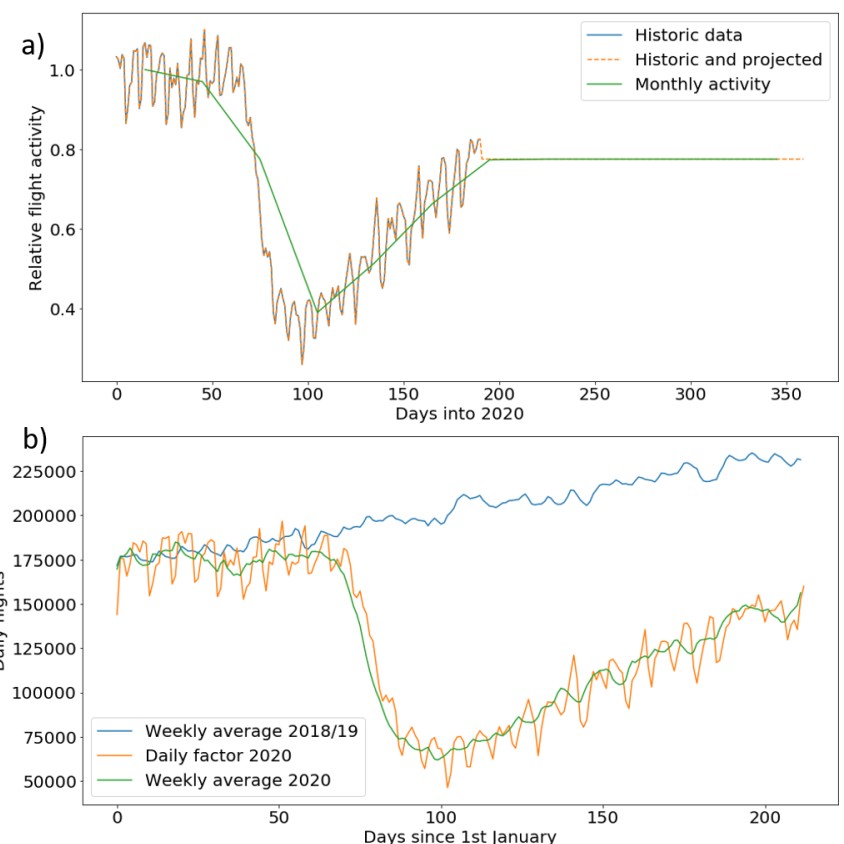

**Figure 4.** Aviation data for a) monthly calculation of activity level - this is normalised to the January data, as previous data from years was not available in open-source format b) workings towards weekly activity level, using closed-source data from previous years too. This approach is used from version 5 onwards for monthly data too.

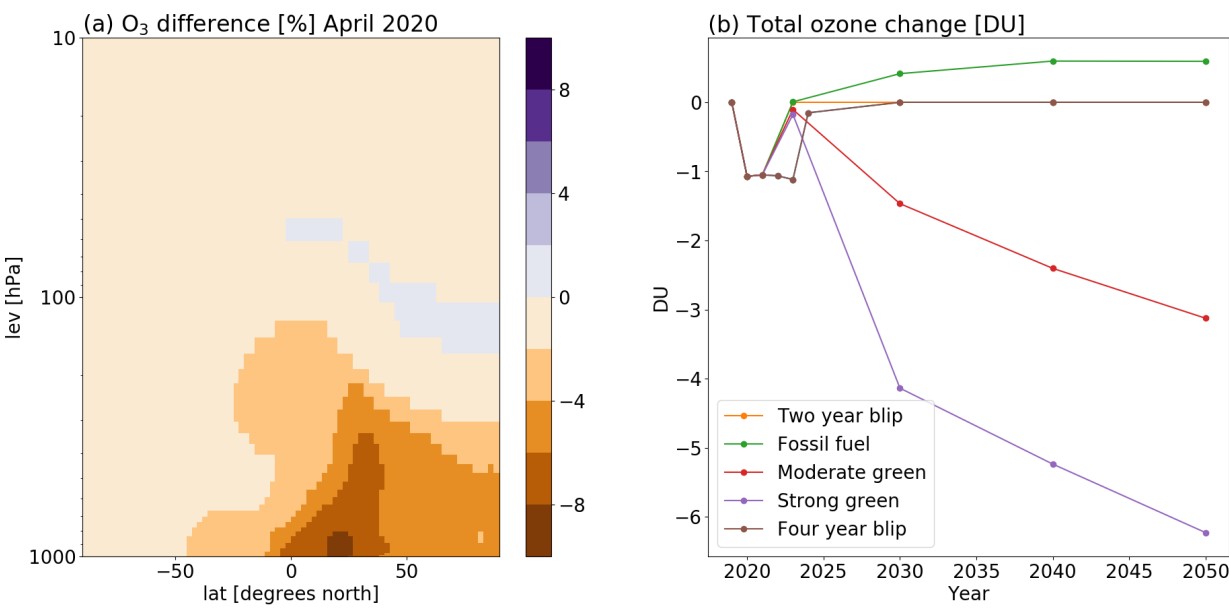

**Figure 5.** a) The relative difference in ozone zonal concentration between the two year blip and baseline in April 2020 (%) in the OsloCTM3. The vertical coordinates in OsloCTM3 are sigma hybrid-pressure levels. The field is plotted for the model levels and indicated by approximate pressure levels on the y-axis. b) The difference in annual total ozone (DU) between the scenarios and the baseline simulations in the OsloCTM3.