# Peer review of "Modifying emission scenario projections to account for the effects of COVID-19: protocol for CovidMIP"

_Geoscientific Model Development, 2020_

## Referee Comment (RC1) · Anonymous Referee #1 · 6 Jan 2021

This paper details the experimental protocol for "COVID-MIP", a series of experiments designed to assess the climatic impact of the COVID-19 pandemic and associated global lockdowns. The study follows from Forster 2020, which produced sectoral estimates of emissions changes due to lockdowns in 2020, along with idealized extrapolations of near- and mid- term recoveries.

The proposed MIP modifies an existing baseline scenario, SSP2-RCP4.5 as simulated by the MESSAGE-GLOBIOM integrated assessment model to produce a transient multi-gas emissions pathway. Experimental protocols are given for a short term experiment to assess climatic effects during the pandemic itself, longer term experiments

to assess the effects of "green" recovery strategies and simple detection/attribution experiments to decompose effects of short lived and long lived forcer perturbations

This MIP is critically necessary at this point in history - by many metrics, the existing scenarios in ScenarioMIP no longer represent the reality we live in. However, exactly because of this importance, the current design falls short of what is required. The paper has not been thoroughly read, with several minor errors, and though the protocol is fairly clear, the design choices neglect to consider the primary uncertainty in the problem at hand: namely, how will the COVID19 pandemic play out?

The protocol assumes a single trajectory for the COVID epidemic, and that trajectory is relatively optimistic (a global elimination of the virus by 2023). As the paper notes, such an event in an integrated assessment model is not even resolved at the default 5 year timescale, and the direct climate impacts are minimal, but it is far from clear that this trajectory is certain.

As noted below, this MIP has one primary role - to produce scenarios which represent and explore the additional uncertainty we now have in global climate projections following the COVID19 pandemic. As such, the focus should be on exploring the uncertainty in the evolution of the pandemic itself and how it modifies the existing scenario framework, which in the worst-case scenario (or even in a middle-of-road scenario) could persist onto the decadal timescale with direct effects on the economy, population, transport and technological progress.

The back-of-envelope projection of the epidemic trajectory was reasonable for the commentary of Forster (2020) as an initial estimate of 2020 emissions impacts - but it is not appropriate for a MIP with a multi-year, multi-center ESM simulation commitment which needs to explicitly explore futures which may be radically different from those considered in ScenarioMIP.

The paper explores 'recovery' strategies of high green investment or an increase in fossil fuel demand - but these are broadly a subset of mitigation responses seen in

ScenarioMIP, but it is unlikely that they will yield major additional insight in the current design. Furthermore, it is not apparent how these recovery scenarios fit into the scenario space as defined in O'Neill (2014).

As such, I recommend a major revision of the protocol to explore the potential degrees of freedom and downstream economic and technological impacts of the pandemic itself.

**Major issues**

*1 - Uncertainty in pandemic parameters*

My major concern with the current protocol is that the parameters of the pandemic itself are represented with certainty, and furthermore the authors have chosen a relatively optimistic scenario - i.e. lockdown measures are projected to decline over a two year period to return to baseline levels in 2023.

These decisions were reasonable in the idealized paper of Forster 2020 - published in the summer of 2020 as a rapid first estimate of emissions effects of the first wave of the pandemic. However, for a MIP protocol which will be primary resource for climate projections in a post-COVID world - the first-order task for this MIP is to explore the representation of uncertainty in the pandemic projection itself, which includes futures where direct COVID impacts persist for a longer timescale.

The lifetime of the direct effects of the pandemic (aside from the recovery plan) are subject to significant uncertainty. Different countries may be subject to significantly different recovery timescales depending of vaccine accessibility (Mullard 2020), and a full elimination by 2023 is optimistic (Ioandinnis 2020)

An important possible avenue to explore is that the direct pandemic effects may persist for a more extended period of time than currently considered. The likely future where there are large portions of the globe with delayed access to vaccines post 2023 would still be subject to regional lockdown measures, travel restrictions and economic effects

beyond the timescale of the experiments in the current design.

There is also the concerning possibility that vaccines will be ineffective against virus mutations. Initial papers classed the likelihood of an endemic scenario to be low (Fernandez 2020), but not impossible, and the emergence of potentially vaccine-resistant strains (Tegally 2020), and more aggressive strains (BMJ 2020) raises at least the possibility that COVID19 variants could remain in circulation for an indefinite future. Assessing the impacts of these futures requires at least the discussion of a completely different type of climate scenario in which global economic norms are radically altered from baseline assumptions on a longer timescale.

The authors need to consider a scenario-based approach to the pandemic evolution (see, for example - McKibben 2020, though now somewhat outdated). At the least, two scenarios for the disease itself are necessary, but ideally more. The existing scenario is an optimistic, global complete recovery by 2023. I suggest that the authors consider the uncertainty in the parameters of the pandemic - with appropriate consideration of a heterogenous regional timescales, with realistic estimates of regional vaccination rates, together with downstream economic, technological and social effects. Furthermore, the potential for an endemic scenario where the direct effects of the pandemic persist onto the decadal timescale should at least be considered.

*2 - Green recovery self-consistency*

The current version considers two "green" recovery scenarios and one fossil fuel intensive scenario in contrast to the baseline SSP2-RCP45. It is unclear in the current version how these fit into the wider scenario framework (O'Neill 2020), where, for example, an aggressive low carbon investment plan is more plausible under SSP1.

But more significantly, it's not clear what these experiments add to the literature in the current MIP framework. The pandemic, as simulated in the current design, is not a significant perturbation to emissions for long term climate targets. As the authors note, a 2020-2023 pandemic would not even be resolvable in the 5 year timestep of most

IAMs.

As such, the pandemic as represented in this projection is insignificant in terms of forcing targets, and the recovery projections duplicate the existing scenario framework, with transient pathways with rapid emission reductions relative to the SSP2-45 baseline (such as SSP1-RCP19 or SSP1-RCP26) or transient pathways with increased fossil fuel growth (such as SSP3-RCP7 or SSP5-RCP85).

It is unclear how GCM simulations of the recovery pathways outlined here add to this literature, given that the scenarios presented here are significantly more idealized than the more complete socio-economic storylines provided for SSP-RCP combinations in the ScenarioMIP project, and that the current paper does not attempt to resolve how the COVID epidemic itself could cause irreversible changes to global economic or technological projections.

**Minor Issues**

line 54: using SSP2-RCP45 is not self-evidently middle-of-the road. The authors should provide more context on why this scenario is in line with current policy

line 21 spelling "unprecedented"

line 100 - "interpolation between the effects of lockdown and the baseline behaviour, so does not need to be interpolated" - sentence is confusing, rewrite.

line 151- spelling "iterations"

line 209 and 129 - spelling "simulations"

line 234 - spelling "minimum"

line 239 - spelling "continuation"

line 299 - spelling "emissions"

line 299, 302 - spelling "simulations"

**References**

Forster, P.M., Forster, H.I., Evans, M.J., Gidden, M.J., Jones, C.D., Keller, C.A., Lamboll, R.D., Le Quéré, C., Rogelj, J., Rosen, D. and Schleussner, C.F., 2020. Current and future global climate impacts resulting from COVID-19. Nature Climate Change, 10(10), pp.913-919.

O'Neill, Brian C., Elmar Kriegler, Keywan Riahi, Kristie L. Ebi, Stephane Hallegatte, Timothy R. Carter, Ritu Mathur, and Detlef P. van Vuuren. "A new scenario framework for climate change research: the concept of shared socioeconomic pathways." Climatic change 122, no. 3 (2014): 387-400. Mullard, Asher. "How COVID vaccines are being divvied up around the world." Nature (2020).

FernaÌ_Ąndez, A., 2020. COVID-19 Evolution in the Post-Vaccination Phase: Endemic or Extinct?. ACS Pharmacology  Translational Science.

Tegally, Houriiyah, Eduan Wilkinson, Marta Giovanetti, Arash Iranzadeh, Vagner Fonseca, Jennifer Giandhari, Deelan Doolabh et al. "Emergence and rapid spread of a new severe acute respiratory syndrome-related coronavirus 2 (SARS-CoV-2) lineage with multiple spike mutations in South Africa." medRxiv (2020).

Ioannidis, John PA. "Global perspective of COVID‐19 epidemiology for a full‐cycle pandemic." European journal of clinical investigation 50, no. 12 (2020): e13423.

McKibbin, Warwick, and Roshen Fernando. The global macroeconomic impacts of COVID-19: Seven scenarios. Centre for Applied Macroeconomic Analysis, Crawford School of Public Policy, The Australian National University, 2020.

Covid-19: What have we learnt about the new variant in the UK? BMJ 2020; 371 doi: https://doi.org/10.1136/bmj.m4944 (Published 23 December 2020), British Medical Journal 2020;371:m4944

---

## Referee Comment (RC2) · Anonymous Referee #2 · 15 Jan 2021

This paper includes the documentation of the use of the Forster et al. data and a new analysis of flight data pertinent to the current conditions in the SSP2.45 data. This includes emissions and concentrations, including ozone and aerosols. The paper is rather straightforward, and serves mostly as a documentation of the COVID-mip protocol. I only have minor comments.

1) There are too many typos in the document (line 138, line 209, lines 281-284, line 299 (2 typos), line 310, ...) which give me the sense that this paper was hastily put together and that the primary authors did not bother re-reading before submitting.

2) Line 32-34: When I read that I was actually quite excited to review this paper. I feel
this paper is far from such a demonstration. It uses the Forster paper for the most part (which IS such a demonstration) and does very little to bring new data for nowcasting analysis. I would reduce the enthusiasm stated here.

3) Line 46: this list of species also include ozone precursors, not just "aerosols and precursors"

4) Lines 87-88: what is the justification for that choice?

5) Line 102: where is the information necessary for interpolation at the daily data? To which sectors does this apply? Is there a consideration of the weekend effect? Who are "certain groups"?

6) Lines 114-115 "We will assume that no changes occured to these sectors" What is the rationale for this assumption? Clearly that is not going to be representative of the real world since I am expecting that solvent industries were affected by COVID.

7) Line 139-140: "This is assumed to be globally uniform and the same across all altitudes". Why? Don't you have all the necessary information from the flight tracking?

8) Line 142: which "one project"?

9) Line 147: word missing "This produces a rather than actual daily factor". What is "everything" in "hence weekly averages are taken of everything"?

10) Line 155: Is it COVID-MIP or Covid-MIP? Be consistent.

11) Line 206: correct spelling of COVID

12) Line 199-201: what is the reason for this sentence. It seems relatively uninformative (why do we need to learn about nudging here?).

13) Sections 7.1 and 7.2 might be more useful presented in a table.

14) Line 237: what is the rational for picking "strong green" as the highest priority?

15) Line 257: CO is not an aerosol precursor, but it is an ozone precursor. So there is
an inconsistency in the protocol if ozone is kept as in SSP2.45.

16) Line 284: Do you mean the diagnostics as in the ScenarioMIP SSP245 simulations?

17) Section 7.4 is rather un-informative. What is the purpose of listing a few variables of interest? This could be replaced by a list of interesting angles that the authors feel justify the need for a COVID-mip.

---

## Short Comment (SC1) · 16 Jan 2021

**Robin Lamboll**

rlamboll@imperial.ac.uk

Received and published: 16 January 2021

article
**COVID19 Methods Reviewer 1 Informal reply**

R Lamboll

January 2021

I thank reviewer 1 for their time and thought in reviewing this document, and will respond more completely in time. Below I will informally address the major concerns to establish what might reasonably resolve them.

**1 Uncertainty in pandemic parameters**

The recommendation to explore multiple scenarios is well-made. We will add an optional scenario based on an extension of the two-year blip to a four-year blip, similarly followed by a one-year return to baseline, that can be included in the second tier of simulations. More complex modelling that accounts for the length of time for vaccines to roll out in different regions is a good suggestion but not realistically tractable in time for the first version of the MIP, which aims to get results published in time for inclusion in AR6.

We will change the text to emphasise that we do not attempt to model the time for the virus to be eliminated/habituated to, but simply for lockdowns to stop interfering with productivity. The data used in the first instance from this MIP (v4) is that collected up
until July. Subsequent data has been collected in the process of making v5, and has shown that the globally-averaged lockdown effects have been slightly less severe than anticipated (currently we are at around a 9% reduction in CO2, whereas we had projected around an 11% reduction in CO2). And while it is far too early to say when the last lockdown will be imposed for most countries, it is clear that even during times of acute lockdown we do not see emissions reductions of a similar intensity to that during the initial lockdown in April 2020, a trend that seems likely to continue as management techniques are developed and an increasing fraction of the population have either been exposed or vaccinated with some variant of the virus, which should confer at least partial protection. Now being one year into the pandemic, the two-year blip scenario thus remains valid as a plausible, mildly pessimistic case of recovery, as it was designed. Our technique produces estimates similar to but slightly more pessimistic in terms of emissions reductions than other techniques such as used by Carbon Monitor. We agree with the reviewer that adding a 'worst case' scenario of extended global infection is valuable, and will include such a scenario in our revisions. Due to the timescales of analysis involved, it is given Tier 2 priority, and will thus form part of the COVID-MIP protocol with analysis expected over the year to come.

**2 Green recovery self-consistency**

We will expand table 2 describing the origin of the scenarios. As the reviewer remarks, the strong decarbonisation is indeed dependent on transferring from an SSP2 world to one more SSP1-like, and the specific breakdown of emissions is consistent with this (e.g. elevated NH3 emissions in the strong green scenarios compared to baseline). Global consistency of the emissions in an SSP1-like world is assured by applying an appropriate method to infill non-CO2 emissions data in the future projections, which uses linear combinations of worlds consistent with the respective SSPs [1]. The other scenarios (including moderate green recovery) are all based on purely SSP2 worlds.
The specific choice of worlds is to investigate the impacts of green investment at the present moment, rather than to encompass the whole range of possible ways the world could evolve. However, the software developed in the course of this project will allow easy application of the impacts of lockdown to be applied to any scenario, and the open-source software is a feature of this paper.

**3 Bibliography**

[1] Robin D. Lamboll, Zebedee R. J. Nicholls, Jarmo S. Kikstra, Malte Meinshausen, and Joeri Rogelj, Silicone v1.0.0: an open-source Python package for inferring missing emissions data for climate change research, Geoscientific Model Development, 2020.

---

## Author Comment (AC1) · 19 Feb 2021

**Reply to RC2**

R Lamboll

February 2021

Many thanks for your time and your warm review. We will address your minor comments as follows:

1. 'There are too many typos in the document': There are indeed, for which we apologise. This paper was caught up in the AR6 deadline and some segments were more reviewed than others! These typos have been corrected.

2. 'Line 32-34: When I read that I was actually quite excited to review this paper. I feel this paper is far from such a demonstration.': It doesn't introduce any fundamentally new nowcasting methods, but the actual data used is updated and the processing for aviation data in recent versions is significantly different to the original. We have changed the sentence to make clear that it is a new use of nowcasting techniques rather than a new demonstration: 'This paper uses data from near-simultaneous "nowcasting" methods based on open-access data'

3. 'Line 46: this list of species also include ozone precursors': changed to 'aerosols and aerosol and ozone pre-cursors'

4. 'Lines 87-88: what is the justification for that choice?': (referring to AFOLU treatment - reduced in Forster 2020 but not for emissions fields here.) We don't really have specific information for AFOLU in any of the data. Agricultural productivity should not be significantly affected by lockdown, so we don't expect emissions to change much either. While it was hoped that deforestation would reduce in line with mobility, this doesn't seem to be true - if anything the opposite, although it varies by country. We now cite papers to explain this. 'This is due to the finding that global deforestation has not slowed down due to lockdown (cite Saavedra2020, Daly2020), and we expect that that agricultural output will remain broadly consistent with pre-lockdown levels.'

5. 'Where is the information necessary for interpolation at the daily data? To which sectors does this apply? Is there a consideration of the weekend effect? Who are"certain groups"': No additional information is needed to do daily data other than for aviation. In activity data the weekend effect is removed in most source data. In practice the daily data (with weekday effects removed) has only been used so far by us for making diagrams and animations, so we have removed the reference to it here. Weekly data has been used in one study, now published and cited here: 'data with every year from 2015 to 2025 is available, as is weekly data for 2020 used by (cite Gettelman2020)'

6. ' "We will assume that no changes occured to these sectors" What is the rationale for this assumption?': As with AFOLU emissions, we expect there to be a general economic rampdown in the medium term, but no acute relationship between the production of solvents/waste and the degree of lockdown, as these are protected industries. And similar to forestry, the lockdown has also reduced government inspection and oversight of emissions, with a possible positive effect. The net impact of this on emissions is unclear.

7. ' "This is assumed to be globally uniform and the same across all altitudes". Why?

Don't you have all the necessary information from the flight tracking?': No, not for free. The open-source flight tracking now provides more data than last year for free, meaning that the seasonal correction used for 7-day data can be imposed for all situations, but still not regionally disagregated data/density maps as far as I know.

8. 'Which "one project"': Gettelman 2020 (https://doi.org/10.1029/2020GL091805), now published as mentioned above.

9. 'Line 147: word missing "This produces a rather than actual daily factor". What is "everything" in "hence weekly averages are taken of everything"?': The missing word was weekly-averaged. We debated using 'pseudo-daily' for cases where we report the data every day but using weekly averages, but have not done so. This now reads: 'This produces a weekly-averaged rather than actual daily factor, since it is not possible to decouple seasonal/holiday and weekday effects. Using weekly averages both removes the weekday effects and reduces the intrinsic variability in the data.'

10. 'Line 155: Is it COVID-MIP or Covid-MIP? Be consistent.' It should now be CovidMIP always - corrected in several places.

11. 'Line 206: correct spelling of COVID': indeed, corrected.

12. 'Line 199-201: what is the reason for this sentence. It seems relatively uninformative (why do we need to learn about nudging here?).': It's quite a useful technique here and we encourage teams to use it where possible. We'll change 'allowed' to 'preferred' to make this clearer.

13. 'Sections 7.1 and 7.2 might be more useful presented in a table.': good suggestion! We will also include the experiments from 7.3 in this table for one big table of experiments.

14. 'Line 237: what is the rational for picking "strong green" as the highest priority?':
It provides the strongest signal and therefore is most likely to have a robustly
detectable result. This is now explained. 'We place the highest priority (tier 1) on
the strong green stimulus recovery as it will likely have the highest signal.'

15. 'line 257: CO is not an aerosol precursor, but it is an ozone precursor. So there
is an inconsistency in the protocol if ozone is kept as in SSP2.45': it's consistent
with the protocol in DAMIP, which does the same thing. This allows a division be-
tween the impact of aerosols directly and the impact of ozone. The nomenclature
for experiments is a little confusing but people seem to have managed so far.

16. 'Line 284: Do you mean the diagnostics as in the ScenarioMIP SSP245 simula-
tions?': correct. Added ', reported for the ScenarioMIP.'

17. 'Section 7.4 is rather un-informative. What is the purpose of listing a few vari-
ables of interest? This could be replaced by a list of interesting angles that the
authors feel justify the need for a COVID-mip.': We want to create an impres-
sion of where we are going with this investigation (and, now results are already
in publication, can hint at the answers here), but you're right that some more
teleological comments would be useful. We have added: 'This [PM2.5 conc] will
allow us to estimate the global impact of lockdown on health effects.' and 'We
expect this MIP will allow us to estimate the continued relevance of climate pro-
jections that do not include the effects of lockdown. If results significantly deviate
from baseline projections, then the continued relevance of outdated simulations
is questioned; if results are broadly similar, old projections can be used with more
confidence.'

---

## Author Comment (AC2) · 19 Feb 2021

**Reply to reviewer 1**

R Lamboll

February 2021

Many thanks for your time and encouraging feedback. We feel the changes we have made in response to it will strengthen the work. As discussed in the personal note, we appreciate that more work will be needed on several fronts, but present this as documentation for an ongoing project, some results of which have already been accepted for publication.

**1   Major issues**

**1.1   Uncertainty in pandemic parameters**

The main criticism, that we did not present a variety of cases for the initial lockdown impacts, is well-made. We will improve the paper by including an additional '4-year blip' scenario (with one year interpolating back to baseline afterwards). This will provide a reasonable upper bound to the timeframe of the direct impact. Comparing projected and recent historical emissions, we see that the 2-year blip assumed the emissions reductions would persist for longer than they have, but would not seek to add a fasterdecaying pathway because the results of simulations for the MIP already performed indicate that it is hard to detect any long-term impact from the 2-year blip already, so adding more pathways between this and the baseline would likely just waste computer time.

While it is possible that some countries will maintain lockdowns for more than 5 years, this scenario would be better handled by a dedicated IAM team to produce a model that accounts for the length of time for vaccines to roll out in different regions. We can simplistically justify a 4-5 year lockdown (although won't in the paper itself) as a crude upper bound considering that almost all developed nations (including China) and the world on average are already vaccinating their population at over 0.06% per day, which gives around 4 years to give at least one injection to everyone. See https://ourworldindata.org/covid-vaccinations for the latest. We expect some acceleration in rollout as more vaccines are coming online and production increases. These vaccines should provide at least partial protection against different strains, and rolling out booster shots should not have a notable impact on emissions. More radical situations of long-term lockdown where the virus mutates and retains high lethality even after any of the vaccines seem unlikely given the speed and diversity of vaccine development, plus much of the population will have had natural exposure by then. We would probably see some sectors of the economy go back to baseline anyway, even if vehicle use and travel remained suppressed. We have change the text to emphasise that we do not attempt to model the time for the virus to be eliminated/habituated to, but simply for lockdowns to stop interfering with productivity.

This additional scenario obviously involves lots of text being altered across the document, we will not attempt to list all of the changes here.

**1.2  Green recovery self-consistency**

In response to the unclarity over the self-consistency of the SSP-nature of the various green scenarios, we have expanded the table documenting the origin of the scenarios. This was nominally covered in Forster 2020 but in practice a lot of details were omitted. The table detailing the origin of the scenarios has been significantly expanded and citations to the full calculation and the methods used to calculate the values have been added. As detailed in the informal response, you are correct that the strong green recovery involves transitioning to an SSP1-like world. The other scenarios are all based on variation between SSP2 worlds.

The primary motivation of this new set of scenarios is to allow resolution of the impact of a step-change in political behaviour now, rather than gradual trends from the point when the scenarios were constructed. A description of this has been added to the introduction: 'This aims to establish the scope of changes in climate results to be expected from the direct impacts of lockdown, and the potential impact of changes to investment structure resulting from the recovery packages.'

**2  Minor issues**

Spelling mistakes have been corrected.

- 'line 54: using SSP2-RCP45 is not self-evidently middle-of-the road. The authors should provide more context on why this scenario is in line with current policy': We have expanded the section justifying our choice with additional citation as follows: 'This amount of forcing is consistent with the global level of warming implied by countries' current NDC pledges (citing ClimateActionTracker) and has most recently projected values closest to the measured emissions (citing Strandsbjerg

2021).'

- 'line 100 - "interpolation between the effects of lockdown and the baseline behaviour, so does not need to be interpolated" - sentence is confusing, rewrite.': This has been rewritten into two sentences: 'The year 2022 is defined as exactly equaling the value interpolated, month-for-month, between the effects of lockdown and the baseline behaviour. This is the normal default infilling method of climate simulators so explicit values are not usually needed here.'

---

## Author Response (AR2)

**Response to second-round review of CovidMIP**

Robin Lamboll

April 2021

We thank the editor for their commitment to rigorous scientific scrutiny and further opportunity to improve the paper. The first reviewer presented no issues for us to resolve, so we simply thank them for their time and will respond only to the points from reviewer 3 here. We remark that several criticisms from reviewer 3 are the opposite of points raised by reviewer 2 from the previous round of reviews, particularly concerning the level of importance of the task and consequently the computational effort that should be expended answering this question. We address their comments as follows:

- Reviewer comment: "While I do see some value in a rapid response tool to update emissions scenarios, I am rather sceptical as to what the authors are trying to achieve with Covid-MIP.":

  The motivation section of the introduction has been significantly expanded. We highlight the potential for findings concerning regional and precipitation differences, which are not convincingly investigable using simplified models. Beyond global and annual averages, many aspects such as regional changes or seasonal/sub-seasonal or even daily extremes can be examined. It is quite plausible that changes in aerosols may affect clear and cloudy sky environments and day and night-time temperatures differently - all aspects which cannot be gleaned from reduced complexity models. This is also briefly flagged in the abstract now.

  The fact that the MIP most likely will (and so far has, see Jones et al. 2021) produced null results on temperature metrics is reassuring to the climate modelling establishment, but is not a priori obvious and may well not hold across all metrics. For instance, there is a small but distinct risk that a perturbation like COVID-19 would result in a disruption to monsoon circulation. Many previous studies have found a sensitivity of monsoons to changes in emissions of aerosols (e.g. Meehl et al., 2008, doi:10.1175/2007JCLI1777.1; Li et al., 2016, doi:10.1002/2015RG000500; Lau et al., 2017, doi:10.1007/s00382-016-3430-y; Zhao et al., 2019, doi:10.1007/s00382-018-4514-7).

- "this would require nudged simulations and/or a clever comparison to observations so as to "control the weather. While nudged simulations are mentioned in the manuscript, they are optional. In the end, I am concerned that the climate modelling groups participating to Covid-MIP will use a lot of computing time to find out fairly small changes in atmospheric composition and climate. What the authors expect to learn on the climate system needs to be much better demonstrated in the manuscript.":

  We describe nudged experiments as "preferred where models have this capacity". We now highlight this feature in the introduction, as well as the importance of sufficient sample size, as especially valuable. As mentioned above, the possibility of seeing precipitation and regional patterns now receives more attention, which nudged models might underrepresent.

  These experiments are computationally much cheaper to run than a normal SSP scenario since they lasts only 30 years and use part of a previously computed run for the initial conditions. The people who had to pay for the computer time have been convinced of their value and the initial results have already been published!

- "This may require a prior quantification of the changes expected before embarking the community in a new MIP.":

  Studies doing this are cited in the paper, including Forster et al. 2020, although using older data (as well as Gettelman et al 2021, which uses a more complex nudged approach). They are, as might be expected, rather small but not obviously undetectable by full GCM models, and at a size where the difference between the simplicity of the simple climate models and the more complex models is likely to be relevant. We now explain that simple climate models are generally bad at capturing regional and precipitation changes, hence the need for a more complete analysis.

  Initial results from other studies, such as Fyfe et al. (doi:10.1126/sciadv.abf7133) also provided guidance during the formation of CovidMIP planning. Their results showed that while global temperature response was indeed

very small, changes in other aspects of the climate system merited further investigation with GCMs. The process of planning spanned many modelling groups, who all showed substantial interest in the activity and the experiments were designed accordingly to be as accessible as possible regards the cost and complication of implementation.

- "The manuscript relies heavily on Forster et al. (2020) and I had to go back to this article to understand the content of this manuscript.":

  We have now greatly extended the introduction to cover the key points of that paper, and expanded the scenario description to explain more about how scenarios were designed.

- "I tend to think IAMs are basically useless to predict the impact of the recovery packages on emissions. In this context the green stimulus scenarios put forward in the manuscript look like wishful thinking, and I do not see their usefulness beyond existing SSP scenarios. I would be happy to be contradicted but I have not seen a convincing argumentation in the manuscript":

  IAMs do not predict the impact of recovery packages (or anything else - they deal only in projections), but they aim to make projections circumscribing the likely probability space of the impacts. The first thing this set of scenarios offers above existing scenarios is that it diverges only in the future, rather than from 2015. We now highlight this more in the text. We expand on the descriptions of how these scenarios were devised to motivate their use. It is also correct that the scenarios examined in this exercise can mostly be considered linear combinations of the existing SSPs, but the advantage of them is that they have a clear narrative link to the events of COVID-19, whereas if we were to simply run the SSPs harmonised to past emissions values we would a) have to wait until the precise values were actually known; b) have no way to link the differences in future behaviour to choices at the moment, since the differences between SSP scenarios is nontrivial; and c) have to handle a discontinuity in the derivative at the end of harmonisation conflated with the impact of lockdown. This discontinuity point is discussed in the next point too.

- "In a sense SSP scenarios used for CMIP6 are already outdated, especially the low ones. The transition between past and future emissions is also poorly represented in these scenarios, with some continuity in emissions but not in the rate of change of the emissions. Maybe the work of the authors can go some way into that direction.":

  To clarify, this point is a criticism of the existing SSP database, not of our work. We agree with this point - it is unrealistic to assume that very strong mitigation action will begin in the earliest modelled timeperiod, and this makes some models in the SSP database age very rapidly. It is in part to avoid this sort of mistake that we always interpolate back to baseline after the end of the lockdown effects (usually in 2023) before starting a new trajectory. In general we do not expect short-term shocks to the economy to result in long-term deviations to emissions. We now refer to recent papers by Le Quere et al. and Gillingham et al. to demonstrate this point and motivate the distinction between long-term and short-term behaviour in our models.

- "Note that there are several initiatives to monitor emissions in near-real-time, it would be interesting to know how the work by the authors compare with others.":

  Work on this front is being investigated by other groups, e.g. the EGU meeting report by Pelletier et al. (https://doi.org/10.5194/egusphere-egu21-16450), indicating that our results based on the Forster method are comparable to the also-unpublished Doumbia results (https://doi.org/10.5194/essd-2020-348). In the cases where something approaching ground truth data have been analysed so far, our results appear to be slightly more accurate, although we will not state this in the paper.

- "In conclusion, the work presented by Lamboll et al is insufficiently motivated. It makes the whole manuscript a little unclear and it is hard to judge if the methodologies chosen to correct the emissions based on various proxies are sound or not for the objectives, given that the objectives themselves are poorly described / not convincing. I recommend rejection.":

  As described above we have more clearly motivated our work, and the take up by 12 models/300 simulations described in Jones et al., 2021 (doi:10.1029/2020GL091883) shows that the wide ESM community agree on the value of this exercise. This anticipated engagement from the community was known in advance and contributed to the development of the protocol described in our manuscript. As with all MIPs not all the final uses are known in advance – we had enough reason to perform the simulations to justify proceeding, but now there is a huge wealth of data available for wide and open analysis. It will stimulate interactions between scientists and scenario modellers. These groups can jointly look at forcings, feedbacks and impacts on atmospheric composition and air quality, resulting in further studies and enhanced experimental design as we learn more about the longer-term impacts of the pandemic on socio-economic behaviour.